# Breast Cancer: Impact of New Treatments?

**DOI:** 10.3390/cancers15082205

**Published:** 2023-04-08

**Authors:** Guy A. Storme

**Affiliations:** Department Radiation Oncology, UZ Brussel, Laarbeeklaan 101, 1090 Brussels, Belgium; guy.storme@telenet.be

**Keywords:** breast cancer, drugs, prevention, screening, genetics, treatment, population data

## Abstract

**Simple Summary:**

This paper states that although numerous new drugs are available for breast cancer, no population benefit is shown from the year 2000 on. It urges the oncological community to revise the “drug approach strategy” in daily practice and pay more attention to early detection and prevention.

**Abstract:**

Background: Breast cancer treatment has seen tremendous progress since the early 1980s, with the first findings of new chemotherapy and hormone therapies. Screening started in the same period. Methods: A review of population data (SEER and the literature) shows an increase in recurrence-free survival until 2000 and it stagnates afterwards. Results: Over the period 1980-2000, the 15% survival gain was presented by pharma as a contribution of new molecules. The contribution of screening during that same period was not implemented by them, although screening has been accepted as a routine procedure in the States since the 1980s and everywhere else since 2000. Conclusions: Interpretation of breast cancer outcome has largely focused on drugs, whereas other factors, such as screening, prevention, biologics, and genetics, were largely neglected. More attention should now be paid to examining the strategy based on realistic global data.

## 1. Introduction

Clinicians and researchers alike raise the question: what is the impact of current treatments on cancer outcome? The most frequent tumor overall (67.2/100.000), and more specifically in women of all races (126.0/100.000), is breast cancer [1]. The incidence increases by 0.42% and 3.45% in the age groups 30–39 and 60–69, respectively [2]. Within the vast volume of breast cancer survivorship intervention research, systematic-review-level research is unevenly distributed, siloed, and with significant gaps in key domains. The most important of these domains is outcome, as it is a major determinant of our attitude toward new approaches. Treatment has been reported on over the years as a good news show, with better outcomes always ascribed to the impact of new available drugs, underestimating cancer screening. The latter lead to the detection of smaller tumors (lower T stage) and, naturally, to better outcomes [3] SEER data covering 27% of all Americans treated for breast cancer show hardly any gain in outcome expressed as years of survival, from the year 2000 onwards [1] (see Figure 1). This is in contrast with the gain rising from 75% to 90.2% since the start of the registration in 1975. The latter period (1975 to 2000) coincides not only with the approval of eight drugs by the Food and Drug Administration (FDA), but also with the initiation of breast cancer screening in the early 1980s [4,5]. Within the same period, reports on results of adjuvant treatments with CMF [6] and hormones [7] were available showing a benefit to survival. Outcome analysis can be performed from several viewpoints: prevention, detection and screening, genetic testing, and treatment.

## 2. Prevention

Prevention is beneficial for the health of the population, but it comes with a price. The number of cancer patients, subject to further evaluation and treatment, is expected to increase between 2012 and 2025 by 37%. Prevention could target pathobiological mechanisms of cancer-related conditions including obesity and diabetes, bearing a higher metabolic cancer risk; the immune system putatively interfering with the progression of precancerous conditions; and possible protumoral side effects of preventive agents shifting balances in the tumor ecosystem.

Body mass index (BMI) reveals that overweight and obesity are associated with worse outcomes in breast cancer patients. For all cancers combined, a BMI of 40 was associated with a 52% (for men) and 62% (for women) increase in mortality as compared to normal weight. For breast cancer, BMIs of 18.5–24.9, 25.0–29.9, 30.0–34.9, 35.0–39.9 and ≥40.0 have relative cancer risks of 1.00, 1.34, 1.63, 1.70, and 2.12 (*p* = 0.001), respectively [8]. Overweight and obese patients were diagnosed later (age at diagnosis: 51.7 ± 12.0 years) than normo-ponderal ones [9]. Breast-adipose-tissue-derived mesenchymal stromal/stem cells (bASCs) were recently identified as crucial components of the tumor microenvironment (TME). Breast-cancer-associated bASCs foster malignancy of breast cancer cells by induction of epithelial–mesenchymal transition and by activation of stemness-associated genes. Mechanistic insight showed how obesity affects the phenotype of bASCs in the TME, and highlights the molecular changes inside breast cancer cells upon interaction with cancer-educated bASCs [10]. Adipose tissue of obese individuals produces inflammatory cytokines and other mediators, creating an environment that promotes cancer invasion and metastasis [11,12]. A case–control study reported that overweight and obesity were associated with higher IBC (Inflammatory Breast Cancer) risk in premenopausal and postmenopausal women (odds ratio [OR], 3.7795% CI, 2.00–7.08) [13]. With mounting evidence linking obesity to a variety of cancers, research focusing on the interaction of adipose tissue and cancer has begun to unravel the intriguing but complex multi-lateral communication between the various players. With breast cancer being one of the first cancer types to show a positive relationship between obesity and breast cancer incidence and prognosis in postmenopausal women, the review focused on the paracrine and endocrine roles of adipose tissue in breast cancer [14].

Focusing on prevention to reduce breast cancer incidence will likely require both a population-based approach to reducing exposure to modifiable risk factors and a precision prevention approach to identifying women at increased risk and targeting them for specific interventions, such as risk-reducing medication. The inclusion of newer factors, such as polygenic risk and mammographic density, can give an idea about individual women’s breast cancer risk. A widespread implementation of evidence-based risk medications is available, but overall implementation remains a challenge [15]. An example is HPV vaccination, which has a well-documented impact on cervix cancer. HPV influences prognosis in cervix cancer and others including head and neck cancers; it has been perceived in many other cancers including breast [16,17]. More research on vaccination is challenging as adjuvant active specific immunotherapy (ASI), with an autologous tumor cell-BCG vaccine combined with surgical resection, was more beneficial than resection alone in stage II and III colon cancers. Recurrence-free survival was significantly longer with ASI (42% risk reduction for recurrence or death [0–68], *p* = 0·032) and there was a trend towards improved overall survival [18]. The reason for why this approach has not been implemented further or taken over by industry is not known.

## 3. Socioeconomics

Socioeconomic conditions that contribute to higher cancer rates in specific populations warrant the analysis of different environmental exposures and the development of policies that might limit these exposures and identify interventions to prevent cancer in people at higher risk. For breast cancer survivors, return to work (RTW) is important from an economic, societal, and personal perspective. One year post surgery, 57% of survivors worked the same and 22% worked reduced working time compared to pre-diagnosis. Impaired RTW was associated with depressive symptoms, arm morbidity, persisting physical fatigue, lower education, and younger age. Cessation of work after breast cancer is associated with worse QoL (Quality of Life) [19].

Racial factors linked or not to socio-economic circumstances play a role in breast cancer outcome. Five-year survival was worse for Black compared to White women in each socioeconomic quartile with 5-year survival hazard ratios of 1.33, 1.23, and 1.20 (*p* < 0.01) in the lowest, second, and third quartile, respectively [20]. The Eurocare 4 database reports striking differences in 5-year breast cancer survival between the Modena (86.9%) and Salerno (71.9) regions in Italy, for unknown reasons [21].

## 4. Medication Impact

Current users of hormone replacement therapy (HRT) have a twice higher risk of developing breast cancer than never users (OR = 2.48; 95% CI = 1.32 to 4.6). For women taking HRT for more than 5 years, the risk is almost three times higher (OR = 2.77; 95% CI = 1.11 to 6.91) [22]. Compared with women who had never used hormonal contraception, the relative risk of breast cancer among all current and recent users was 1.20 (95% CI = 1.14 to 1.26). This risk increased from 1.09 (95% CI = 0.96 to 1.23) with less than 1 year of use to 1.38 (95% CI = 1.26 to 1.51) with more than 10 years of use (*p* = 0.002). After discontinuation of hormonal contraception, the risk of breast cancer remained higher (1.0 to 1.6) after use for 5 years or more [23].

Prevention by medication is exemplified by tamoxifen, reducing breast cancer incidence by more than 30%. In a recent paper, Rowan and colleagues report that across three tamoxifen placebo-controlled prevention trials (N = 23,360), started almost 30 years ago, although there were 226 fewer breast cancer cases, there were nine more deaths from breast cancer in the tamoxifen groups [24]. As endocrine-targeted agents commonly prevent these cancers, widespread implementation of current prevention strategies may not reduce deaths from breast cancer. They suggest that re-examination of breast cancer risk reduction strategies and clinical practice should be considered [25]. Tamoxifen and raloxifene, and the aromatase inhibitors exemestane and anastrozole, all significantly reduce the incidence of primarily ER-positive and (PR)-positive breast cancers, but again no reduction in breast cancer deaths has been observed. So far, using an aromatase inhibitor rather than tamoxifen in premenopausal women receiving ovarian suppression reduces the risk of breast cancer recurrence. Longer follow ups are needed to assess the impact on breast cancer mortality [26].

Whether or not ER- positive patients benefit from adjuvant chemotherapy is an open question. In the TAILORx trial, 6907 women were randomly assigned to endocrine therapy alone or endocrine therapy plus chemotherapy with 3399 and 3312 women available for an analysis according to the randomized treatment assignments in both arms, respectively. After a median follow-up of 90 months, the difference in invasive DFS—the main study end point—met the prespecified noninferiority criterion (*p* > 0.10 for a test of no difference after 835 events had occurred), suggesting the noninferiority of endocrine therapy compared with endocrine therapy plus chemotherapy. In this population, the 9-year invasive DFS rate was 83.3% for endocrine therapy alone and 84.3% for endocrine therapy plus chemotherapy (hazard ratio [HR], 1.08; 95% confidence interval [CI], 0.94–1.24; *p* = 0.26) [27].

At the St Gallen expert consensus in 2007, three categories were recognized: highly endocrine responsive, partially endocrine responsive, and endocrine nonresponsive. The panel accepted HER2 positivity to assign trastuzumab but noted that adjuvant trastuzumab has only been evaluated in conjunction with chemotherapy. Chemotherapy is typically given with or before trastuzumab in patients with HER2-positive disease, and it may be used in patients with endocrine-responsive disease when endocrine therapy alone is insufficient [28].

Metastasis is the major cause of poor outcomes. Cancer cells escape from the primary tumor, invade the blood vessels, and home in on distant organs such as the lung, liver, and bone [29]; there, these disseminated tumor cells (DTCs) may remain dormant, wake up at an unpredictable time, and start to develop into deadly clinical metastases [30]. DTCs are not very sensitive to chemotherapy, as exemplified by the observations of Stephan Braun’s team. Of fifty-nine patients, 29 (49.2%) and 26 (44.1%) presented with cytokeratin (CK)-positive cells, presumably tumor cells, in the bone marrow. After chemotherapy, less than half of the previously CK-positive patients (14 of 29 patients; 48.3%) had a CK-negative bone marrow, and 11 (36; 7%) of 30 previously CK-negative patients turned CK positive [31]. The percentage of patients with metastatic synchronic disease hardly changed during the years 1973, 1998, and 2008 (7.64%, 5.53%, and 6.82%, respectively), in contrast with metachronous disease (23.3%, 6.33%, and 5.31%, respectively. This could be explained by the availability of more accurate diagnostic tools, such as bone scan, CT scan, and last but not least, PET. The latter was approved in 2002 for breast cancer [32].

The overall survival of breast cancer might be sufficiently explained by earlier stage and less evidence of any major impact of systemic treatment [33]

## 5. Screening

There is no doubt that screening has an impact on breast cancer outcome as it detects smaller tumors. However, mammography and echography show no benefit over physical examination [34]; they rather have a harmful effect, especially due to overdiagnosis [35]. Novel screening techniques such as MRI, lacking harmful ionizing radiation, do not show an overall benefit over mammography [36]. A randomized trial in the Netherlands revealed that MRI screening detected cancers at an earlier stage than mammography in women with familial cancer [37]. The lower number of late-stage cancers identified in incident rounds might reduce the use of adjuvant chemotherapy and decrease breast-cancer-related mortality [38]. However, the advantages of the MRI screening approach might be at the cost of more false-positive results, especially at high breast density [39]. Pre-NST (neoadjuvant systemic therapy) MRI detected additional lesions in 31% of patients and resulted in more extensive surgery in 26% of these patients, including 5% contralateral surgeries [40]. One wonders about the impact of breast cancer screening, as calculations across different studies showed that on average it took 10.7 years (4.4 to 21.6) before one death from breast cancer was prevented for 1000 women screened [41].

## 6. Genetic Impact

Mapping the genes responsible for inherited breast cancer may allow the identification of early lesions that are critical for the development of breast cancer in the general population. Chromosome 17q21 harbors a gene for inherited susceptibility to breast cancer in families with early onset disease [42]. Adjuvant chemotherapy guided by a 21-gene expression assay in breast cancer suggest that the 21-gene assay may identify up to 85% of women with early breast cancer who can be spared adjuvant chemotherapy [27]. Recently, AI for making treatment decisions was implemented on the basis of pre-therapy features, including tumor mutational and copy number landscapes, tumor proliferation, immune infiltration, and T cell dysfunction and exclusion. Combining these features into a multiomic machine learning model predicted a pathological complete response in an external validation cohort [43]. Whether or not those approaches will be effective remains an open question given the detection of driver mutations in at least 40 cancer genes and 73 different combinations of mutated cancer genes. The results highlight the substantial genetic diversity underlying this common disease [44]. We would like to recall here that the histological grading phenotype, recently refined and objectivated by virtual microscopy, encompasses genetic diversity and remains a significant predictor of breast cancer outcome [45]. 

## 7. Treatment Impact

The statement that the increase in the 5-year relative survival for breast cancer from 76.3% to 90.2% between 1980 and 2000 (Figure 1) is due to the impact of treatment and screening is an oversimplification. To quote Jayasekera and Mandelblatt JS, “…. Decisions about the cost effectiveness of methods to reduce breast cancer mortality will remain difficult without comprehensive analyses across the full spectrum from prevention to screening and treatment, including consideration of implementation costs. Confidence in results might also be facilitated by evaluating the potential for bias in methods, findings, and interpretation of findings” [46]. In the later period (2000–2016) covered by the SEER table (Figure 1), survival hardly changed (range 90.2–91.4%), despite the FDA’s approval of 28 new drugs between 2000 and 2010 and considering that we have only data of 5-year survival until 2016. Can we conclude that these drugs are not effective? Certainly not, but how can we explain the lack of correlation between the survival results shown by the SEER data and the promising messages from the clinical trials? Such trials suffer from obvious weakness, most notably bias in the selection criteria zooming in on a lower-risk study population, which is not representative of the larger population covered by the SEER data [47].

Similar questions emerge from Belgian data, where survival increased between 2004 and 2017 by 6% up to 91% with mainly eight drug combinations. At the same time, screening was nationally introduced, resulting in 6% smaller tumors [48]. In the Netherlands, the proportion of patients with stage I also increased from 28 to 41% and stage II decreased from 51 to 33%, resulting in 91% 5-year relative survival [49] with the same drugs available.

More recent studies implicating new drugs have been designed on the basis of novel molecular technology. The cost effectiveness of this strategy is more difficult to evaluate than established care because of high initial pricing, limited data on efficacy or long-term effects, and potential for bias in the methods and findings as a result of industry-related conflicts of interest. So far, the number coded for these new molecules as Biological/ Immunology is 47, and on the website of cancer.org 25 refer to breast cancer. Looking to the SEER Interactive Antineoplastic Drugs Database (seer.cancer.gov), for breast drugs, which have at least some indication of referring to breast cancer, we find 248 single drugs and 393 regimens available.

If the latter drugs were effective, we should at least see better outcomes within the first 1 to 4 years. A recent analysis of 124 FDA drugs approved for cancer between 2003 and 2021 showed only a small benefit of 2.8 months (IQR, 1.97–4.60 months) for overall survival (OS) and 3.30 months (IQR, 1.50–5.58 months) for progression-free survival (PFS) [50].

It is often unclear how long and at what dose an antitumor drug should be administered for an optimal effect at minimal toxicity. It is usually not known how the drug is best combined with surgery, radiotherapy, and other pharmacological agents. These uncertainties complicate the decision making of doctors and their patients. The European Medical Agency (EMA) and the European Organization of Research and Treatment of Cancer (EORTC) collaborating in the Cancer Medical Forum (CMF) plans to tackle this problem [51]. Now, the FDA guidance addresses considerations for the design and analysis of externally controlled trials to study the effectiveness and safety of drugs. Although various sources of data can serve as the control arm in an externally controlled trial, the FDA guidance focuses on the use of patient-level data from other clinical trials or from real-world (RWD) sources, such as registries, electronic health records (EHRs), and medical claims [52]. Do new drugs improve the outcome for stage IV breast cancer? A recent report shows median survival times between 2.8 and 3.1 over the period 2004–2017 [53].

The explosion of new molecules with different approaches and running trials (N = 706) tested in the USA on breast cancer [54], most of which show a small benefit in selected populations, present a major task to the CMF. In the meantime, physicians should be urged to use only the treatments they handle with expertise.

## 8. Conclusions

As a conclusion, we can only say that when interpreting breast cancer relative survival, we should look at the entire population rather than individual cases or treatment populations, and consider the impact of stage migration, which influences outcome. Prevention is undoubtedly important, and education in the young population should be prioritized.

## Figures and Tables

**Figure 1 cancers-15-02205-f001:**
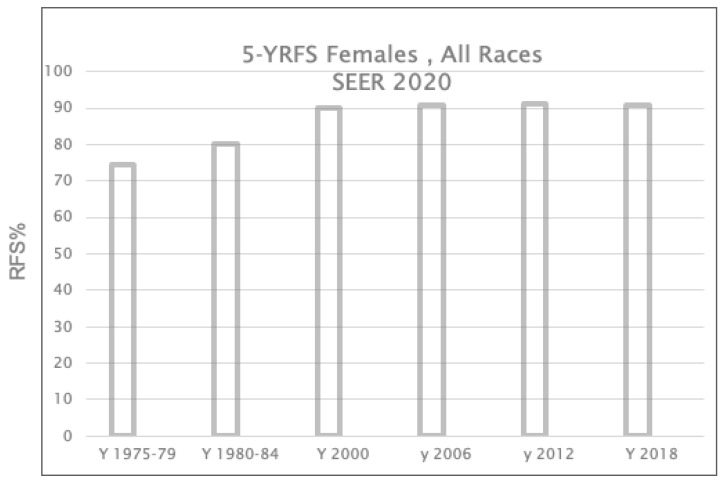
SEER 2020 5-year Relative Free Survival (RFS)% by year of diagnosis; all races; females; 5-year RFS.

## Data Availability

The data presented in this study are available in this article.

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
