# Peer review of "Breast Cancer: Impact of New Treatments?"

_cancers, 2023, doi:10.3390/cancers15082205_

Round 1
Reviewer 1 Report
This manuscript “Breast Cancer: Impact of new treatments?”, calls attention to early prevention and detection of breast cancer. The starting point of this review is interesting. Multiple aspects of breast cancer prevention research are described, including risk factors such as overweight and obesity, socioeconomic conditions, and hormones.
However, the structure of the manuscript is not clear. It is recommended to add some subsection headings to divide the article into different sections to help the readers grasp the content of the article. Now, there is a subtitle, maybe, “Breast Cancer: Outcome related to screening or treatment?”. The abstract need to be rewrite.
In current form I do not feel the article is suitable to be accepted.
Reviewer 2 Report
- The manuscript was poorly prepared, and the structure was not clear. The manuscript should be separated into several sections to make it easier to read, such as background, methods, results, and discussion. If the author intended to review the recent advances in breast cancer treatment, subtitles are needed to make the manuscript more clear.
- The first line of the manuscript was a bold subtitle: Breast Cancer: Outcome related to screening or treatment? But I didn't see any other subtitles.
- The resolution of the figures was poor and hard to read.
- I suggest the author carefully proofread the manuscript to avoid manifest errors. Such as, on page 2 of 8, paragraph 3, the second sentence is 'cells in obesity and breast cancer: spatial regulation and function'. Why is this sentence here? And the word 'cells' be capitalized?
- Some sentences were hard to read, and the English writing needed to be improved.
- The methods in abstract was not method.
- 5-y was used several times over the manuscript; should it be spelt out as 5-year?
Reviewer 3 Report
This manuscript presents a shrewd and insightful review which discusses the importance of early detection and prevention rather than new treatments in generating the recent improvement in breast cancer outcomes. The manuscript is informative and interesting. However, there are several points that might be considered to strengthen the manuscript. Major comments: 1. In the upper part of Figure 1, it might be better to show changes in survival rates graphically rather than numerically. 2. Regarding the description of BMI in Page 2, it has been reported that being overweight or obese increases the risk of breast cancer, especially in postmenopausal women. It is suggested that higher estrogen levels due to the activity of aromatase in adipose tissue are associated with the development of estrogen receptor-positive breast cancer. Please also consider this point. 3. In Figure 2, please provide an explanation for the bars such as blue, yellow, etc. in the bar chart. 4. The authors showed that there has been hardly any gain in outcome expressed as years of survival from the year 2000 despite the approval of new drugs (Figure 1). Moreover, they stated that screening has an impact on breast cancer outcome because of the detection of smaller tumors. This is true and this reviewer understands the author’s point. On the other hand, new treatments may improve the prognosis of metastatic/recurrent breast cancer. Are there any data that show survival time after recurrence or Stage IV breast cancer before and after 2000? Minor comments: Page 4, line 10: “estrogen positive patients” should be “estrogen receptor-positive patients”. Page 6, in the last sentence of the text: “not de drug combination” might be “not the drug combination”.Author Response
please see the attachment.

Round 2
Reviewer 2 Report
1. Proofreading is required to remove minor errors, such as editing marks, from the manuscript (the strikethroughs on lines 60, 133, and 204).
2. Lines 107–108 duplicate a sentence from lines 96–97; why is this sentence here?
Author Response
I thanks Rev 2 to have reread attentively the text and I apologize I didn't see this.
- Correct about strikethroughs line 133 need and 204 of should be adapted, but I can't since I only can read the text, even I can't duplicate it. Line 60 is not available in the text I received: it goes from 59 to 70
- Lines 107–108 duplicate a sentence from lines 96–97; why is this sentence here? Complete correct this last sentence line 107 should be omitted (For breast cancer survivors...perspective